# TOWARDS UNDERSTANDING THE CAUSE OF ERROR IN FEW-SHOT LEARNING

## ABSTRACT

Few-Shot Learning (FSL) is a challenging task of recognizing novel classes from scarce labeled samples. Many existing researches focus on learning good representations that generalize well to new categories. However, given low-data regime, the restricting factors of performance on novel classes has not been well studied. In this paper, our objective is to understand the cause of error in few-shot classification, as well as exploring the upper limit of error rate. We first introduce and derive a theoretical upper bound of error rate which is constrained to 1) *linear separability* in the learned embedding space and 2) *discrepancy* of task-specific and task-independent classifier. Quantitative experiment is conducted and results show that the error in FSL is dominantly caused by classifier discrepancy. We further propose a simple method to confirm our theoretical analysis and observation. The method adds a constraint to reduce classifier discrepancy so as to lower the upper bound of error rate. Experiments on three benchmarks with different base learners verify the effectiveness of our method. It shows that decreasing classifier discrepancy can consistently achieve improvements in most cases.

## 1 INTRODUCTION

Learning novel concepts from few samples is one of the most important ability in human cognition system (Chen et al. (2018); Dhillon et al. (2019); Wang et al. (2020)). By contrast, massive achievements of modern artificial intelligent systems are dependent upon lots of data and annotation which are hard to acquire in many scenarios. Blocked by the difficulty in obtaining large labeled datasets, community shows more interests in developing algorithms with high data-efficiency. It is so-called few-shot learning that learns to generalize well to new categories with scarce labeled samples (Sung et al. (2018); Vinyals et al. (2016)). Existing methods deal with few-shot learning in the general framework of meta-learning where a base learner is developed and optimized across different episodes (or tasks). Episodes are formed in a $N$-way $K$-shot fashion where $K$ support samples per class are available for training. The overall objective is enabling the base learner to exploit on base classes and to transfer learnt knowledge to recognize novel classes with few support data. Since training and evaluation are performed on different tasks, the base learner holds different task-specific classifiers that depend on data sampling.

In general, classification model has two components: feature extractor and classifier (Simonyan & Zisserman (2015); He et al. (2016); Zagoruyko & Komodakis (2016)). Most approaches of few-shot learning exploit from according perspectives: learning a good embedding and finding a right base learner. Rethinking-FSC (Tian et al. (2020)) demonstrates that a good learned embedding space can be more effective than many sophisticated meta-learning algorithms. It argues for the performance on meta set where embeddings are learnt in supervised or self-supervised way. Goldblum et al. (2020) reveal the importance of feature clustering in few-shot learning. Since classifier performance is sample-dependent especially in one-shot scenario, variance of feature is expected to be small so as to retain good performance. It shows that classifier performance is not stable across different tasks. MetaOptNet (Lee et al. (2019)) and R2-D2 (Bertinetto et al. (2018)) explore training and optimization routines for linear classifier, enabling good few-shot performance through simple base learner. These literatures develop specific algorithms from the aspects of learning good representation or optimizing base learner. Most recent methods use linear classifier as base learner, so we also consider linear model in this paper. To our best knowledge, there has been little research focusing on how the

two components (aka feature representation and classifier) respectively influence the performance on novel classes in FSL.

In this paper, we introduce an upper bound of error rate in few-shot learning, indicating that the error comes from two aspects: 1) *linear separability* in the embedding space and 2) *classifier discrepancy* between task-specific and task-independent classifiers. The ideal classifier is viewed to be task-independent since its performance is not sample-dependent (Goldblum et al. (2020)). To quantitively estimate each term, an experiment is performed where we use error rate of supervised classification tasks on novel classes to measure feature separability, and use disagreement of results obtained from different classifiers to measure discrepancy. It comes to an interesting observation that features learned through simple methods are sufficiently discriminative and the error mainly comes from classifier discrepancy. Based on our observation and theoretical analysis, we propose a simple method of reducing classifier discrepancy so as to boost few-shot performance. Experiments on three benchmarks are conducted to empirically verify our theory. Results on different datasets with various base learners show consistent improvements, supporting our finding and theory in few-shot learning.

The main contributions of this paper are:

1. The upper bound of error rate on novel classes is theoretically analyzed. From derived equations we figure out that the error in FSL is caused by *linear separability* in the feature space and *discrepancy* between task-specific and task-independent classifiers.

2. Quantitative experiments are conducted to verify the theoretical analysis. Results show that the error is dominantly caused by classifier discrepancy.

3. Based on the theoretical analysis and the experiment results, a constraint is proposed to reduce classifier discrepancy so as to decrease the upper bound of error rate in FSL.

4. Further experiments on mini-ImageNet, tiered-ImageNet and CIFAR-FS confirm the effectiveness of the proposed method. It shows that decreasing classifier discrepancy can consistently achieve improvements in most cases.

## 2 RELATED WORK

**Algorithms of Few-Shot Learning** Prototypical Network (Snell et al. (2017)) is a classical algorithm for its simplicity and effectiveness, which performs few-shot classification by nearest-prototype matching. Since class prototype is the mean of features, the linear separability in feature space has direct impact on classification results. Performance of following series of prototype based methods (Allen et al. (2019); Liu et al. (2020)) is also limited by feature separability. Different with these methods using nearest-neighbor classifier, Bertinetto et al. (2018) adopt ridge regression and logistic regression as base learner. Similarly, Lee et al. (2019) use classical linear classifier SVM in few-shot learning to learn representations. Simple linear classifier shows competitive performance and in this paper, we use linear classifier in measuring linear separability and classifier discrepancy.

**Theoretical Analysis of Few-Shot Learning** Cao et al. (2019) introduce a bound for accuracy of Prototypical Network (Snell et al. (2017)), demonstrating that the intrinsic dimension of the embedding function's output space varies with the number of shots. They further propose a method to overcome the negative impact of mismatched shots in meta-train and meta-test stages. As in (Liu et al. (2020)), they give a lower bound for accuracy cosine similarity based prototypical network. Two key factors: intra-class bias and cross-class bias are theoretically formulated. We also analyze theoretical bounds in few-shot learning. Theory in this paper does not focus on specific algorithm like Prototypical Network but on general scenarios, from the perspective of feature separability and classifier discrepancy.

**Theoretical Analysis of Domain Adaptation**: Methods of domain adaptation (Ben-David et al. (2007; 2010); Ganin & Lempitsky (2015)) solve the problem of how to train a classifier on source domain and guarantee the classifier performs well on target domain. A classifier's target error is bound by its source error and the divergence between the two domains in (Ben-David et al. (2010)). They utilize $\mathcal{H}$-divergence and $\mathcal{H}\Delta\mathcal{H}$-divergence to measure discrepancy between two domains. $\mathcal{H}\Delta\mathcal{H}$-divergence can be computed from finite unlabeled data, allowing us to directly estimate the

error of a source-trained classifier on the target domain. Inspired by their work, we also use $\mathcal{H}\Delta\mathcal{H}$-divergence to measure discrepancy between sets on novel classes and base classes.

# 3 BACKGROUND

## 3.1 PROBLEM SETUP

The common setup of few-shot learning used in this paper is described below. A space of class is divided into two parts: base classes $C^{base}$ and novel classes $C^{novel}$ where $C^{base} \cap C^{novel} = \emptyset$. Dataset $D^{base}$ of base classes is used for model training and the model is evaluated on dataset $D^{novel}$ whose samples belong to unseen classes during training. The model is composed of a feature extractor $F$ and a classifier $h$. In few-shot learning, we usually consider $N$-way $K$-shot $Q$-query tasks $T$. In task $\tau_i = (D_i^s, D_i^q, h)$, the support set $D_i^s$ includes $K$ data $x \in \mathbb{R}^d$ per class and its true label $y \in \{c_1, ..., c_N\}$. The goal is to predict labels for query data in $D_i^q$ given $D_i^s$. In this paper, we use error rate on novel classes to evaluate the few-shot performance of a trained model. The error rate is formulated as:

$$\epsilon_{novel} = E[\epsilon_\tau] = \frac{1}{M \times Q} \sum_i^M \sum_j^Q \mathbb{1}(h(F(x_{i,j}))! = y_{i,j}) \tag{1}$$

where $M$ is the number of sampled tasks $\tau_i \sim T^{novel}$. $\mathbb{1}(\cdot)$ is indicator function.

## 3.2 DISTRIBUTION DIVERGENCE

We adopt following concepts to explore the cause of error in few-shot scenarios.

**Definition 1** Given a set $D = \{(x_1, y_1), ..., (x_m, y_m)\}$ where $x_i \in \mathcal{X}$ and $y_i \in \mathcal{Y}$, for any mappings $h_1, h_2 \in \mathcal{X}$, **disagreement** is defined in Eqn. 2 to measure the difference of these two mappings.

$$dis(h_1, h_2) = P_{x \sim \mathcal{D}_\mathcal{X}}(h_1(x) \neq h_2(x)) \tag{2}$$

**Definition 2** Given a domain $\mathcal{X}$ with $\mathcal{D}_1$ and $\mathcal{D}_2$ probability distributions over $\mathcal{X}$, let $\mathcal{H}$ be a hypothesis class on $\mathcal{X}$ and denote by $I(h)$ the set for which $h \in \mathcal{H}$ is the characteristic function; that is, $x \in I(h) \Leftrightarrow h(x) = 1$. $\mathcal{H}$ **divergence** between $\mathcal{D}_1$ and $\mathcal{D}_2$ is

$$d_\mathcal{H}(\mathcal{D}_1, \mathcal{D}_2) = 2sup_{h \in \mathcal{H}}|Pr_{\mathcal{D}_1}[I(h)] - Pr_{\mathcal{D}_2}[I(h)]| \tag{3}$$

**Definition 3** For hypotheses $h, h' \in \mathcal{H}$, the symmetric difference hypothesis space $\mathcal{H}\Delta\mathcal{H}$ is the set of hypotheses $g \in \mathcal{H}\Delta\mathcal{H} \Leftrightarrow g(x) = h(x) \oplus h'(x)$ where $\oplus$ is the XOR function. $\mathcal{H}\Delta\mathcal{H}$ **divergence** over distributions is defined as following:

$$d_{\mathcal{H}\Delta\mathcal{H}}(\mathcal{D}_1, \mathcal{D}_2) = 2sup_{h \in \mathcal{H}}|Pr_{x \sim \mathcal{D}_1}[h(x) \neq h'(x)] - Pr_{x \sim \mathcal{D}_2}[h(x) \neq h'(x)]| \tag{4}$$

# 4 PROPOSED METHOD

## 4.1 MEASURING CLASSIFICATION PERFORMANCE ON NOVEL CLASSES

A classification model generally consists of two parts: feature extractor and classifier. Hence, the model holds two expectations: 1) Extracted features are expected to be discriminative for classification (Snell et al. (2017); Lee et al. (2019)). 2) Classifier is supposed to be stable concerning different tasks (Cao et al. (2019); Liu et al. (2020)). Consider linear classifier in this paper, we investigate from **linear separability** and **classifier stability** to measure classification performance on novel classes. To quantitively estimate these two terms, we design experiments using a base model with ResNet-12 backbone and FC (fully-connected) layer classifier. The base model is first trained on base classes in supervised way. When testing on novel classes, we replace FC layer classifier with ProtoNet (Snell et al. (2017)) and ridge regression (RR) as in paper Ye et al. (2020). The $\epsilon_{novel}(h^*)$(i.e. error rate $\epsilon$ on $C^{novel}$ with classifier $h^*$), is used to approximate and quantify feature separability on novel classes (Eqn. 5).

$$\epsilon_{novel}(h^*) = \frac{1}{N \times Q^*} \sum_i^{N \times Q^*} \mathbb{1}(\hat{y}_i \neq y_i) \tag{5}$$

Table 1: Experiment of *linear separability* and *classifier discrepancy* of 5-way ($N$=5) classification tasks on novel classes.

| Dataset | Classifier | $\epsilon_{novel}(h^*)$ | $dis(h, h^*)$ | |
|---|---|---|---|---|
| | | | 1-shot | 5-shot |
| mini-ImageNet | PN | 11.72% | 39.32% | 16.23% |
| tiered-ImageNet | | 11.28% | 32.65% | 14.60% |
| mini-ImageNet | RR | 6.72% | 38.34% | 19.45% |
| tiered-ImageNet | | 5.97% | 31.10% | 16.94% |

In Eqn. 5, $Q^*$ is the number of all samples of each class $c \in \{c_1, ..., c_N\}$. $\hat{y}_i$ is predicted label and $y_i$ is true label. $h^*$ is trained under supervised way from large set of samples and is used to approximate the expected ideal $N$-way classifier. On the other hand, we use disagreement defined in Eqn. 6 to measure classifier discrepancy.

$$dis(h, h^*) = \frac{1}{N \times Q} \sum_i^{N \times Q} \mathbb{1}(\hat{y}_i \neq \hat{y}_i^*) \tag{6}$$

where $Q$ is the number of query samples. $h$ is task-specific classifier that differentiates among tasks, decided by sampled support data. $h^*$ is task-independent concerning these $N$ classes. Thus, $dis(h, h^*)$ indicates the discrepancy between the task-specific classifiers and the ideal classifier.

Table 1 shows results on two benchmarks: mini-ImageNet and tiered-ImageNet. From Table 1, we can see that $\epsilon_{novel}(h^*)$ is generally lower than $dis(h, h^*)$ in a large margin. For example, 1-shot $dis(h, h^*)$ on mini-ImageNet with RR is up to 38.34% while $\epsilon_{novel}(h^*)$ is 6.72%, which is five times lower. Furthermore, the obvious drop of $dis(h, h^*)$ from 1-shot to 5-shot indicates obvious raising of classifier discrepancy. An interesting conclusion can be drawn from this experiment that *the error on novel classes is dominantly caused by classifier discrepancy in low-data regimes rather than linear separability*. More details about this experiment are presented in the appendix.

### 4.2 UPPER BOUND OF ERROR RATE

In this section, we introduce an upper bound of error rate on novel classes in few-shot learning.

**Proposition 1** *Consider a feature extractor $F$ and a hypothesis space $\mathcal{H}$. Based on triangular inequality, for $h, h^* \in \mathcal{H}$, it follows that:*

$$\epsilon(h; F) \leq \epsilon(h^*; F) + dis(h, h^*; F) \tag{7}$$

$h^*$ is the ideal hypothesis in $\mathcal{H}$, holding that $h^* = \arg\min_{h \in \mathcal{H}} E[\epsilon_\tau(h; F)]$. *Proof* is in the appendix.

In few-shot learning, error rate on novel classes is usually denoted by $\epsilon_{novel} = E[\epsilon_\tau(h; F)]$. Hence, the upper bound is:

$$\epsilon_{novel} \leq E[\epsilon_\tau(h^*; F)] + E[dis_\tau(h, h^*; F)] \tag{8}$$

where $\tau \sim T^{novel}$. From Eqn. 8, boosting few-shot performance can be achieved by minimizing the two terms in right side of above inequality. However, $h^*$ is unavailable when testing on novel classes. In order to connect the performance on novel set and it on base set we consifer following setting.

Consider $N$-way $K$-shot tasks where $N$ is assumed to be same in meta-train and meta-test stages. $h, h'$ are linear classifiers of novel classes and base classes respectively. For classification weights $W_b, W_n \in \mathbb{R}^{N \times d}$ of base and novel classes, there exists a linear transformation matrix $\tilde{W} \in \mathbb{R}^{d \times d}$ that has $W_b = W_n \tilde{W}$. We define the linear transformation between the ideal hypothesis on the novel set and that on the base set as $\Lambda$, $h'^* = \Lambda(h^*)$, $h^* = \Lambda^{-1}(h'^*)$. For query samples $X = \{x_i \in \mathbb{R}^d\}$, predicted results are:

$$h(X; W_n) = X W_n^T = X(W_b \tilde{W}^{-1})^T \tag{9}$$

According to the above analysis we know that performing a transformation on the classifier is equivalent to perform the transformation on the data, $\Lambda(h)(X) = h(\Lambda(X))$.

**Lemma 1** *Let $\mathcal{H}$ be a hypothesis space of VC dimension d. $h^*$, $h'^*$ are ideal hypotheses on $D^{novel}$ and $D^{base}$. There is an ideal hypothesis $\hat{h} = \arg\min_{h \in \mathcal{H}} \epsilon_{novel}(h) + \epsilon_{base}(\Lambda^{-1}(h))$. Then for $h^*$ and $h'^*$:*

$$\epsilon_{novel}(h^*) \leq \epsilon_{base}(h'^*) + \frac{1}{2}d_{\mathcal{H}\Delta\mathcal{H}}(D^{novel}, D^{base}) + \lambda \tag{10}$$

*where $\lambda = \epsilon_{novel}(\hat{h}) + \epsilon_{base}(\Lambda^{-1}(\hat{h}))$ is the combined error of the ideal hypothesis $\hat{h}$.*

**Lemma 2** *For linear hypotheses $h, h' \in \mathcal{H}$ and ideal hypotheses $h^*$ on $D^{novel}$, $h'^*$ on $D^{base}$, there exists:*

$$
\begin{aligned}
& dis(h, h^*; D^{novel}) \\
& \leq dis(h, \Lambda(h'^*); D^{base}) + \frac{1}{2}d_{\mathcal{H}\Delta\mathcal{H}}(D^{novel}, \Lambda(D^{base})) \\
& \leq dis(h', h'^*; D^{base}) + dis(h', \Lambda(h); D^{base}) + \frac{1}{2}d_{\mathcal{H}\Delta\mathcal{H}}(D^{novel}, \Lambda(D^{base}))
\end{aligned}
\tag{11}
$$

*Proofs* of Lemma 1 and Lemma 2 are provided in the appendix. We give the core theory in this paper by plugging Lemma 1 and Lemma 2 into Eqn. 8.

**Theorem 1** *The upper bound of error rate on novel classes in few-shot learning is:*

$$
\begin{aligned}
\epsilon_{novel} \leq & \epsilon_{base}(h'^*) + dis(h', h'^*; D^{base}) \\
& + dis(\Lambda^{-1}(h), h'; D^{base}) + \lambda + \frac{1}{2}d_{\mathcal{H}\Delta\mathcal{H}}(D^{novel}, D^{base}) + \frac{1}{2}d_{\mathcal{H}\Delta\mathcal{H}}(D^{novel}, \Lambda(D^{base}))
\end{aligned}
\tag{12}
$$

Based on theoretical analysis and experiments, we come to several conclusions:

1. In theory, the error rate of few-shot classification is influenced by linear separability of feature representation and classifier discrepancy between task-specific and task-independent classifiers. Experiment results indicate that the main cause of error in few-shot learning is classifier discrepancy.

2. From Theorem 1, we can see that the upper bound of error rate on novel classes is positively related to 1) linear separability on $D^{base}$, 2) classifier discrepancy on $D^{base}$, 3) the combined error and 4) $\mathcal{H}\Delta\mathcal{H}$-divergence of $D^{novel}$ and $D^{base}$ measuring the discrepancy between two domains.

## 4.3 REDUCING CLASSIFIER DISCREPANCY FOR FEW-SHOT LEARNING

Based on our theoretical analysis, we propose a simple method to reduce the upper bound of error rate, boosting few-shot performance by reducing classifier discrepancy.

**Measuring Classifier Discrepancy** It is proved in Sec. 4.2 that reduce error on novel classes can be achieved by improving linear separability and reducing classifier discrepancy. Furthermore, experiment reveals that the cause is the discrepancy between task-specific classifier and ideal classifier. For these reasons, we target to reduce the upper bound by decreasing classifier discrepancy. Discrepancy denoted in Eqn. 6 is non-differentiable so that we propose two measurements of classifier discrepancy to ease gradient propagation in training stage. Since we consider linear classifier in this paper, an intuitive way to reduce classifier discrepancy is to reduce distance between classification weights. Squared euclidean distance can be used as a measurement:

$$dis_{MSE}(h, h^*; W, W^*) = E[\|W - W^*\|_2^2] \tag{13}$$

where $W$ is the weight of task-specific classifier $h$ and $W^*$ is the weight of task-independent classifier $h^*$. $dis_{MSE}$ measures the variance of classification weights. Since task-specific classifier is decided by data sampling, taking data distribution into consideration, we suggest to calculate the difference of logit predicted by various classifiers. Consider commonly used KL divergence to measure the difference of logits:

$$dis_{KL}(h, h^*) = E[KLD(h(D^q; D^s), h^*(D^q))] \tag{14}$$

**Training Policy** Training process of proposed method ***Reducing Classifier Discrepancy (RCD)*** consists two phases. In the first phase, model is trained in conventional supervised way on base classes. Loss function in phase 1 is:

$$L_{sup} = L_{ce}(h(F(x)), y), \text{with } (x, y) \sim D^{base} \tag{15}$$

Table 2: Performance of RCD. 5-way classification accuracies (%) without/with $dis_{MSE}$ constraint. Results in **bold** indicate performance is improved by reducing classifier discrepancy.

| Backbone | Setting | Dataset | PN | | LR | | RR | |
|---|---|---|---|---|---|---|---|---|
| | | | - | $dis_{MSE}$ | - | $dis_{MSE}$ | - | $dis_{MSE}$ |
| ResNet-12 | 1-shot | mini-ImageNet | 60.18 | **61.42** | 61.28 | **61.43** | 62.30 | **63.64** |
| | | tiered-ImageNet | 66.24 | 65.84 | 64.23 | **65.03** | 66.77 | **67.48** |
| | | CIFAR-FS | 67.90 | **71.02** | 66.86 | **70.77** | 67.92 | **70.27** |
| | 5-shot | mini-ImageNet | 79.07 | **80.10** | 77.64 | 77.27 | 79.34 | **80.94** |
| | | tiered-ImageNet | 82.98 | 82.35 | 81.62 | **82.64** | 82.63 | **83.12** |
| | | CIFAR-FS | 84.38 | **86.33** | 83.14 | **85.20** | 82.73 | **85.55** |
| ConvNet-64 | 1-shot | mini-ImageNet | 52.61 | **54.39** | 51.43 | **51.99** | 53.40 | **54.05** |
| | | tiered-ImageNet | 59.71 | **61.92** | 57.44 | **57.60** | 60.11 | **61.27** |
| | | CIFAR-FS | 68.59 | 68.20 | 67.94 | **68.21** | 69.10 | **70.55** |
| | 5-shot | mini-ImageNet | 71.33 | **72.55** | 70.95 | 70.39 | 72.58 | 72.30 |
| | | tiered-ImageNet | 74.18 | **75.06** | 74.39 | **75.01** | 75.35 | **76.22** |
| | | CIFAR-FS | 78.05 | **79.80** | 78.54 | **78.66** | 79.46 | **80.14** |

where $L_{ce}$ is standard cross-entropy loss. The classifier $h^* = \arg\min_h L_{sup}$ obtained in this stage is treated as the ideal classifier, also the task-independent classifier, on $D^{base}$. With fixed $h^*$, we train the model on meta tasks $T^{base}$ with loss in Eqn. 16:

$$L_{meta} = L_{ce}(h(F(x)), y) + \beta\, dis(h, h^*), \text{with } (x, y) \sim T^{base} \qquad (16)$$

The second training procedure aims to reduce classifier discrepancy on base classes. Consequently, the upper bound of $\epsilon_{novel}$ can be decreased as verified in Theorem 1.

Our method shows great flexibility that training in the first phase is free of carefully designing base learners. Moreover, policy in the second phase can generalize to different base learners. Algorithm is shown in the appendix.

## 5 EXPERIMENTS

### 5.1 IMPLEMENTATION DETAILS

**Dataset and Backbone** We conduct experiments on three benchmarks: mini-ImageNet (Vinyals et al. (2016)), tiered-ImageNet (Ren et al. (2018)) and CIFAR-FS (Bertinetto et al. (2018)). ResNet-12 (Lee et al. (2019)) and ConvNet-64 (Snell et al. (2017)) are employed as backbones in this paper. Details about data setting and architectures are shown in the appendix.

**Base Learner** To illustrate the effectiveness of our proposed method, we use three base learners: PN, Ridge Regression (RR) and Logistic Regression (LR) (Bertinetto et al. (2018)). **1)PN** is derived from (Snell et al. (2017)) which finds the nearest prototype based on cosine similarity. Prototype is computed from support samples: $P = norm(\frac{1}{K}\sum_i^K F(x_i))$. Predicted labels of query samples are given by $\hat{Y} = \arg\min_c Cos(P_c, X)$. **2) RR** Classification weight is estimated by $W = (X^T X + \gamma I)^{-1}X^T Y$ where $Y$ is one-hot labels of support samples and $I$ is identity matrix. Prediction of query samples are $\hat{Y} = X \cdot W$. **3) LR** Classification weight in logistic regression is $W = \arg\min_W L_{ce}(D^s, W)$. Query samples are predicted by $\hat{Y} = X \cdot W$. Descriptions are detailed in the appendix.

### 5.2 RESULTS OF RCD

We evaluate the proposed Reducing Classifier Discrepancy (RCD) on three benchmarks, with three base learners and two backbones. Table 2 and Table 3 summarize few-shot results with $dis_{MSE}$ and $dis_{KL}$ be auxiliary loss respectively. Overall, RCD achieves consistent improvements in most cases.

**Auxiliary Loss** As displayed in Eqn. 16, two measurements of classifier discrepancy can be added as auxiliary loss in meta-train stage. We compare the results without and with auxiliary constraints in Table 2 and Table 3. Each second column of three base learners shows few-shot accuracies by reducing classifier discrepancy constrained by $dis_{MSE}$ or $dis_{KL}$. Generally, $dis_{MSE}$ is an useful

Table 3: Performance of RCD. 5-way classification accuracies (%) without/with $dis_{KL}$ constraint. Results in **bold** indicate performance is improved by reducing classifier discrepancy.

| Backbone | Setting | Dataset | PN | | LR | | RR | |
|---|---|---|---|---|---|---|---|---|
| | | | - | $dis_{KL}$ | - | $dis_{KL}$ | - | $dis_{KL}$ |
| ResNet-12 | 1-shot | mini-ImageNet | 60.18 | **63.25** | 61.28 | **62.59** | 62.30 | **64.41** |
| | | tiered-ImageNet | 66.24 | **67.37** | 64.23 | **65.11** | 66.77 | **67.97** |
| | | CIFAR-FS | 67.90 | **71.35** | 66.86 | **70.21** | 67.92 | **71.06** |
| | 5-shot | mini-ImageNet | 79.07 | **80.82** | 77.64 | **78.50** | 79.34 | **80.27** |
| | | tiered-ImageNet | 82.98 | **83.32** | 81.62 | **82.03** | 82.63 | **83.40** |
| | | CIFAR-FS | 84.38 | **86.49** | 83.14 | **85.96** | 82.73 | **85.49** |
| ConvNet-64 | 1-shot | mini-ImageNet | 52.61 | **55.52** | 51.43 | **52.02** | 53.40 | **54.25** |
| | | tiered-ImageNet | 59.71 | **62.94** | 57.44 | **57.97** | 60.11 | **62.04** |
| | | CIFAR-FS | 68.59 | **68.62** | 67.94 | **68.10** | 69.10 | **70.73** |
| | 5-shot | mini-ImageNet | 71.33 | **73.40** | 70.95 | **72.15** | 72.58 | **73.88** |
| | | tiered-ImageNet | 74.18 | **76.23** | 74.39 | **75.09** | 75.35 | **77.63** |
| | | CIFAR-FS | 78.05 | **78.69** | 78.54 | **79.31** | 79.46 | **79.55** |

Table 4: Changes of classifier discrepancy on novel classes. Columns of *Stage 1* show discrepancy on novel classes after the fist conventional training stage. Columns of *Stage 2* show discrepancy on novel classes after the second training stage with auxiliary constraints $dis_{MSE}$ or $dis_{KL}$. Differences are highlighted by green and red respectively. Green/red means reduced/enlarged discrepancy. Backbone in this experiment: ResNet-12.

| RCD | Setting | Dataset | PN | | LR | | RR | |
|---|---|---|---|---|---|---|---|---|
| | | | Stage 1 | Stage 2 | Stage 1 | Stage 2 | Stage 1 | Stage 2 |
| $dis_{MSE}$ | 1-shot | mini-ImageNet | 39.32% | 37.96%(-1.36) | 39.48% | 36.99%(-2.49) | 38.34% | 36.24%(-2.10) |
| | | tiered-ImageNet | 32.65% | 32.82%(+0.17) | 32.21% | 32.01%(-0.20) | 31.10% | 30.28%(-0.82) |
| | | CIFAR-FS | 30.73% | 27.49%(-3.24) | 31.15% | 29.64%(-1.51) | 30.64% | 29.03%(-1.61) |
| | 5-shot | mini-ImageNet | 16.23% | 15.81%(-0.42) | 16.26% | 16.40%(+0.14) | 19.45% | 18.86%(-0.59) |
| | | tiered-ImageNet | 14.60% | 14.76%(+0.16) | 15.92% | 15.64%(-0.28) | 16.94% | 16.54%(-0.40) |
| | | CIFAR-FS | 12.58% | 11.25%(-1.33) | 13.78% | 12.03%(-1.75) | 14.17% | 12.30%(-1.87) |
| $dis_{KL}$ | 1-shot | mini-ImageNet | 39.32% | 33.85%(-5.47) | 39.48% | 34.57%(-4.91) | 38.34% | 33.21%(-5.13) |
| | | tiered-ImageNet | 32.65% | 31.84%(-0.81) | 32.21% | 31.72%(-0.49) | 31.10% | 30.11%(-0.99) |
| | | CIFAR-FS | 30.73% | 26.03%(-4.70) | 31.15% | 27.59%(-3.56) | 30.64% | 27.44%(-3.20) |
| | 5-shot | mini-ImageNet | 16.23% | 15.70%(-0.53) | 16.26% | 16.02%(-0.24) | 19.45% | 16.43%(-3.02) |
| | | tiered-ImageNet | 14.60% | 14.02%(-0.58) | 15.92% | 15.39%(-0.53) | 16.94% | 15.98%(-0.96) |
| | | CIFAR-FS | 12.58% | 10.80%(-1.78) | 13.78% | 12.74%(-1.04) | 14.17% | 12.62%(-1.55) |

constraint which results in improvements up to 2.82% on 1-shot CIFAR-FS with RR. By contrast, $dis_{KL}$ shows superiority in reducing classifier discrepancy that by adding constraint $dis_{KL}$, accuracy is raised in all cases, up to 3.45% on 1-shot CIFAR-FS with PN. Under the same conditions, reducing classifier discrepancy can bring in larger improvements in 1-shot scenarios. For example, on mini-ImageNet with PN, reducing classifier discrepancy through diminishing KL divergence achieves improvement by margins of 2.91% in 1-shot and 2.07% in 5-shot. Experiment results are consistent with our theory. In 1-shot settings, discrepancy is larger due to data scarcity. Thus, RCD makes more obvious increase on 1-shot tasks.

**Base Learner** We argue for the flexibility and generalization of proposed RCD. For verification, we adopt three commonly used linear classifiers as base learner in meta-train stage. Results in Table 2 and Table 3 demonstrate feasibility of our method in improving few-shot performance regardless of specific classifier. It indicates that reducing classifier discrepancy on base set is effective to lower the upper bound of $\epsilon_{novel}$.

**Backbone** In this section, ResNet-12 and ConvNet-64 are used for ablation study. Overall performance on ResNet-12 is prominently better than performance on ConvNet-64. Moreover, RCD makes relatively larger improvements when taking ResNet-12 as architecture. Note that dimension of features extracted by ConvNet-64 is 64 while dimension of ResNet-12 features is 640. Higher dimension indicates larger hypothesis space. That is, classifier in higher-dimension space is less stable so classifier discrepancy is larger. ResNet-12 holds a higher-dimension space where reducing classifier discrepancy can result in more obvious improvements.

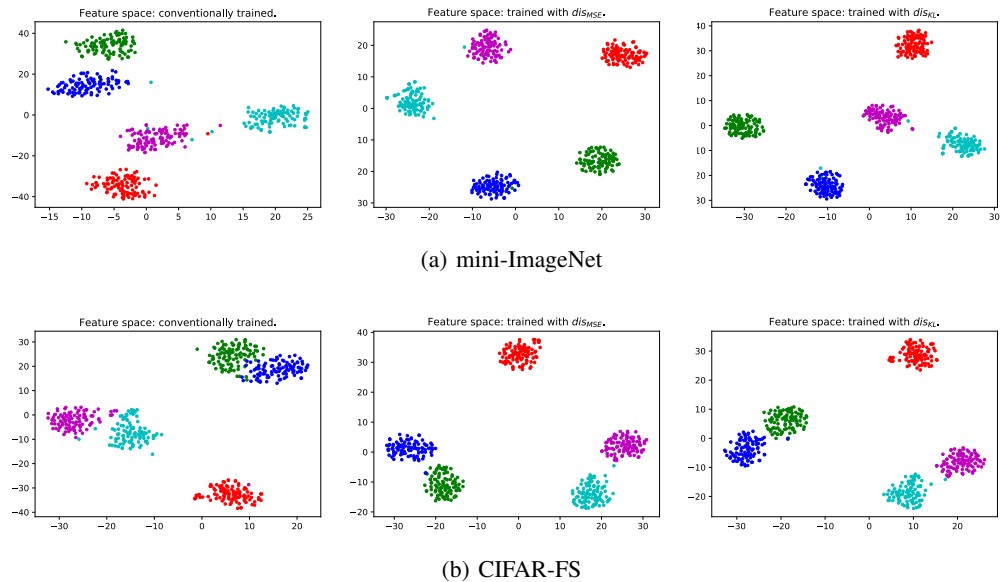

(a) mini-ImageNet

(b) CIFAR-FS

Figure 1: t-SNE visualization of features of novel classes. The first column: feature space is learnt in conventional training. The second and third column: feature space is learnt with $dis_{MSE}$ and $dis_{KL}$. In each dataset, 5 novel classes are randomly sampled. Backbone: ResNet-12. Base learner: PN. Best viewed in color.

## 5.3 CLASSIFIER DISCREPANCY

We compare changes of discrepancy in Table 4 to clearly illustrate the proposed method is effective in reducing classifier discrepancy. Changes are denoted in colors. It can be clearly see that after the second training stage with constraints $dis_{MSE}$ and $dis_{KL}$, classifier discrepancy is reduced in nearly all settings, especially on mini-ImageNet and CIFAR-FS. Downward trend of classifier discrepancy is positively correlated to decreasing the upper bound of error rate. Results in Table 4 are consistent with accuracy increment in Table 2 and Table 3, further proving the rationality of our proposed theory.

## 5.4 VISUALIZATION

T-SNE visualization (Maaten & Hinton (2008)) is provided in Fig. 1 to give an intuitive understanding of our method. In Fig. 1, figures in the first column display the distribution of features that trained in conventional way. Figures in latter columns visualize features trained with proposed constraints $dis_{MSE}$ and $dis_{KL}$. We can see that features within same classes cluster more tightly and the boundaries among different classes become more clear. Our method enables larger separability in feature space.

## 6 CONCLUSION

In this paper, we theoretically analyze the upper bound of error rate on novel classes in few-shot learning. We derive that the upper bound is decided by feature separability and classifier discrepancy. Furthermore, the observation shows that classification error is mainly caused by classifier discrepancy in few-shot scenarios. Based on our observation and theory, we propose a simple method to lower the upper bound of classification error by reducing classifier discrepancy. Two differentiable discrepancy measurements are proposed as auxiliary constraints in our method RCD which is feasible on different base learners. To verify the feasibility and generalization of proposed RCD, comprehensive experiments are conducted on three few-shot benchmarks with three base learners. Experiment results powerfully prove that RCD is effective to reduce classifier discrepancy and consequently lower the upper bound of error rate in few-shot learning.

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
