# OpenReview forum: " Towards Understanding the Cause of Error in Few-Shot Learning"
_ICLR.cc/2021/Conference — Reject_

### Official Review · AnonReviewer3 · 2020-10-19
**Proofs unclear, hard to follow**

**Rating:** 4
**Confidence:** 5

**Review:**

Summary: This paper seeks to understand theoretically the current bottleneck in few-shot learning and address it with a new way of training the embedding. The authors find that the key issue in few-shot learning is not the separability of the novel classes but the discrepancy between classifiers trained on large datasets and few-shot classifiers. They then propose a way to reducee this discrepancy.

Pros:
+ The analysis seems to be along a worthwhile direction
+ the proposed modfications look straightforward and simple.

Cons:
- The theoretical parts of the paper are very poorly written and peppered with unexplained notations and incorrect claims. Some examples:
-- Definitions (3) and (4) seem to implicitly assume that $h$ is boolean, which makes no sense in the multiclass few-shot learning setup.
-- Equations (5) and (6) seem to change notations between them, with $h^*(x)$ being referrred to as $\hat{y}$  in (5) but as $\hat{y}^*$ in (6)
-- The statement of Proposition 1 mentions triangle inequality without any indication of there being norms or metric spaces where a triangle inequality might apply.
-- The paragraph before Equation (9) assumes that the classifiers for novel classes are a linear transformation of the classifiers for the novel classes, and that this linear transformation is invertible (9). Neither assumption is ever proved.
-- The proof for Lemma 1 introduces terms such as $\epsilon_N(h^*, h'^*)$ that are never defined. This term seems to relate to $dis(h, h')$ introduced in the main paper for measuring the discrepancy between two classifiers, but that makes no sense since $h^*$ and $h'^*$ operate on completely different label spaces.
(There are probably more; I ran out of steam at this point).
- Comparisons to state-of-the-art in few-shot learning is missing.

---

> ### Author Response · Authors · 2020-11-24
> **Response to Reviewer#3**
>
> Thank you for reading our paper carefully and making constructive comments.
>
> 1. We assume that you refer to Eq. 3 (Definition 2) and Eq. 4 (Definition 3) in this question (since we do not have Definition 4 in the main paper). Definition 2 and Definition 3 are given in [1]. We use binary classification in discussion, however, multi-class classification can be formulated as the problem of multiple binary classification. Thus, it can generalize to multi-class few-shot learning.
>
> 2. Thank you for pointing out the typos and we will fix them in the revised version.
>
> 3. The derivation of Proposition 1 is given in Appendix 2.1. We will delete the statement of triangle inequality in the revised version since the derivation is irrelevant to it.
>
> 4. We will remove the notation of invertible transformation since it is not used in the following derivation.
>
> 5. Thank you for your question. $\epsilon_{N}(h^*, h'^*)$ refers to the classifier discrepancy, denoted by $dis_{novel}(h^*, h'^*)$ in the main paper. We will give consistent notations in our follow-up works.
>
> 6. Our main contribution is deriving a theory bound in few-shot learning rather than achieving a state-of-the-art result. We focus on improving performance by additional constraint, aka reducing classifier discrepancy during training. It is proper to compare with baselines, columns denoted by '-' in Table 2 and Table 3, to show the effectiveness of our method. We achieve consistent improvements over baselines and the results are align with our theory.
>
>
>    [1] Ben-David S, Blitzer J, Crammer K, et al. A theory of learning from different domains. Machine learning, 2010, 79(1-2): 151-175.

---

### Official Review · AnonReviewer1 · 2020-10-28
**The upper bound seems loose.**

**Rating:** 4
**Confidence:** 4

**Review:**

-----------------------------------
Summary:
This paper analyzes the upper bound of error rate on novel classes in few-shot learning theoretically. It derives that the upper bound is decided by feature separability and classifier discrepancy and shows that classification error is mainly caused by classifier discrepancy in few-shot scenarios. In addition, this paper proposes a new method to lower the upper bound of classification error by reducing classifier discrepancy.

-----------------------------------
Pros:
1. The author proposes an upper bound of the error rate on few-shot learning.
2. The classifier discrepancy loss is further introduced on top of the traditional methods to reduce the error rate on novel classes.
3. Extensive experiments and a visualization of the feature space are provided to demonstrate the proposed method's effectiveness.

-----------------------------------
Cons:
1. In Eq.5, isn't it supposed to be (\hat{y_i^{\ast}}\neq y_i)? Otherwise, it contradicts Eq.6.
2. Table 1 should include \epsilon_novel(h). Otherwise, it is hard to support the statement that the error on novel classes is dominantly caused by classifier discrepancy. For a fixed \epsilon_novel(h^{\ast}), a large dis(h, h^{\ast}) does not necessarily imply a large \epsilon_novel(h).
3. Eq.10 is too loose. As the base dataset and the novel dataset have no intersection, it is not feasible to approximate the classifier on the novel dataset with a classifier trained on the base dataset by domain adaptation. The marginal distribution won't be matched, let alone the optimal joint error.
4. The linear transformation \tidle{W} does not appear in the final objective Eq.16; thus, the reviewer doesn't know whether it is used during the training. Besides, by comparing Eq.16 and the proposed upper bound Eq.12, it seems that only the first two terms of the upper bound are optimized, which makes the entire method not persuasive enough.
5. Linearity assumption for classifiers makes no sense as a simple ReLU function will introduce non-linearity, and the reviewer doesn't think it is necessary according to the final objective.

-----------------------------------
Reasons for Score:

The reviewer rates 4/10 because the third concern is critical.

---

> ### Author Response · Authors · 2020-11-24
> **Response to Reviewer#1**
>
> Thank you for reading our paper carefully and making constructive comments.
>
> 1. Thank you for pointing out the typos and we will fix them as suggested.
>
> 2. Actually, values of $\epsilon_{novel}(h)$ are implicitly listed in Table 2 and 3. Columns noted by '-' show accuracies which are same to $1-\epsilon_{novel}(h)$.
>
> | Dataset          |  Classifier  |  $\epsilon_{novel}(h^*)$  |  $\epsilon_{novel}(h)$ 1-shot |  $dis(h, h^*)$ 1-shot  |  $\epsilon_{novel}(h)$ 5-shot  |  $dis(h, h^*)$  5-shot  |
> | :-------------------: | :--------------: | :-------------------------: | :--------------------------------: | :--------------: | :--------------------------------: | :---------------: |
> | mini-ImageNet         | PN               | 11\.72%                     | 39\.82%                            | 39\.32%          | 20\.93%                            | 16\.23%           |
> | tiered-ImageNet       | PN               | 11\.28%                     | 33\.76%                            | 32\.65%          | 17\.02%                            | 14\.60%           |
> | mini-ImageNet         | RR               | 6\.72%                      | 37\.7%                             | 38\.24%          | 20\.66%                            | 19\.45%           |
> | tiered-ImageNet       | RR               | 5\.79%                      | 33\.23%                            | 31\.10%          | 17\.37%                            | 16\.94%           |
>
> 3. Thank you very much for pointing out this problem. We find it exactly loose. Therefore, we will solve this problem in subsequent improvements.
>
> 4. (1) We do not target to find an explicit transformation $\tilde{W}$ of classification weights between novel and base classes. Since the ideal classifier on novel classes is unavailable, the transformation $\tilde{W}$ is hard to explicitly optimized. (2) There are some terms regarding the distribution of novel classes in Eq. 12. These terms are unavailable and non-differentiable during training. Therefore, we focus on reducing the differentiable classifier discrepancy in Eq. 12.
>
> 5. (1) In this paper, we decompose the model into a feature extractor and a classifier. The feature extractor is generally a neural network with convolution layers, and the classifier is generally linear. Popular classifiers such as PN [1] and SVM [2] do not have non-linear layers like ReLU. (2) The assumption of linear transferablity of linear classifiers is a necessary step to derivate the final objecitve.
>
>
>    [1] Snell J, Swersky K, Zemel R. Prototypical networks for few-shot learning. In NIPS 2017.
>
>    [2] Lee K, Maji S, Ravichandran A, et al. Meta-learning with differentiable convex optimization. In CVPR 2019.

---

### Official Review · AnonReviewer4 · 2020-10-30
**Upper-bonding the error is useful, but doubts about the proposed method**

**Rating:** 5
**Confidence:** 4

**Review:**

Summary:

This paper aims to understand the cause of error in few-shot classification. The authors are particularly interested in the upper-bound of the error rate, which they break down into linear separability in the feature space of the meta-train classes and classifier discrepancy on the meta-train classes, among other terms. Empirical results show that the latter is the dominant term. After identifying this, the authors propose a method, Reducing Classifier Discrepancy, to reduce the classifier discrepancy, lowering the upper-bound of the error rate. Empirical results show the benefits of the proposed method on three few-shot datasets.

Pros:
1. This work investigates the cause of error in few-shot classification and finds an upper-bound for it.
2. The proposed method, Reducing Classifier Discrepancy, is simple. Empirical results on three few-shot datasets show its effectiveness.

Cons:
1. Through experimentation, the authors find that classifier discrepancy dominates the few-shot classification error. This could be due to the dataset being used for the experiments. This might not hold true in a cross-domain dataset, such as Meta-Dataset [1]. Can results be provided on such a dataset?
2. The proposed method, Reducing Classifier Discrepancy, has two training phases. There have been works [2, 3] that show that after the first phase of conventional supervised training, the model does very well on few-shot tasks. How does the performance after the first phase compare to the that of the proposed method? Is the gain in the proposed method coming from the model being trained longer?
3. Additionally, the classifier discrepancy loss forces better clustering of samples in the meta-train dataset. Can this not be achieved by training for longer or using better hyper-parameters?

Clarifications:
1. In Equation 9 of the appendix, is the expansion of the discrepancy between \Lambda(h) and h^{\prime *} required for the result?

Notes:
1. It is hard to follow the equations. Additionally, the notation changes going from the main paper to the appendix. I would suggest cleaning that up for better readability.
2. Equation 12 seems to have a typo - \Lambda^{-1} instead of \Lambda.

[1] Mengye Ren et al. Meta-Learning for Semi-Supervised Few-Shot Classification.
[2] Wei-Yu Chen at al. A Closer Look at Few-shot Classification.
[3] Guneet S. Dhillon et al. A Baseline for Few-Shot Image Classification.

---

> ### Author Response · Authors · 2020-11-24
> **Response to Reviewer#4**
>
> Thank you for reading our paper carefully and making constructive comments.
>
> 1. Thank you for your suggestion about experiments on cross-domain datasets. Indeed, this work focused on standard few-shot setting and didn't consider the cross-domain problem. It is possible that the classifier discrepancy is not necessarily the dominant factor in the cross-domain setting. Models might have poor seperability because of huge gap among datasets. Due to limited time, we will conduct more discussions on this issue in the follow-up works.
> 2. (1) It is unnecessarily for supervised learning tasks to benefit from longer training time, as discussed in [1]. In addition, as the training time increases, the model is more likely to be over-fitted on base classes in few-shot learning and can not generalize well to novel classes. We will conduct detailed experiments in the follow-up work to verify it. (2) Hyper-parameter and constraint are two aspects. We do not focus on carefully selecting hyper-parameters.
> 3. Eq. 9 in Appendix gives the relationship between the classifier discrepancy $dis_{base}$ on base classes and the classifier discrepancy $dis_{novel}$ on novel classes. By Eq. 9, we can obtain low discrepancy $dis_{novel}$ by reducing $dis_{base}$ which is computable during training.
> 4. We will unify the notations in the main paper and appendix for better readability.
> 5. Thank you for pointing out the typos, and we will correct them in the follow-up works.
>
> [1] Tian Y, Sun C, Poole B, et al. What makes for good views for contrastive learning. In NeurIPS 2020.

---

### Official Review · AnonReviewer2 · 2020-10-30
**This paper shows mathematically and experimentally that reducing classifier discrepancy leads to better results.**

**Rating:** 6
**Confidence:** 5

**Review:**

I have not had the time to write a detailed review but I have gone through the paper carefully.
Motivation
The paper is motivated well and presents an interesting mathematical derivation that drives the subsequent approach based on reducing classifier discrepancy. The manuscript has a number of typos and grammatical errors for example "consider" is spelled as "consifer" and "first" as "fist" among others. Despite those problems, the logical flow is good so the paper is acceptably readable.

Method
The mathematical results is derived for linear classifiers which lend themselves to simple analysis. The authors then say that experimentally classifier discrepancy is the more significant factor, and propose a technique based on reducing such discrepancy. The authors test their approach on various datasets.

Results
The authors' results are decent. The improvements are small but consistent which lends some credence to their approach. I find their results modest.

Quality, clarity, originality and significance
The clarity of this paper is acceptable and will be better if the typos and grammatical errors are corrected. The originality of the paper as well as the quality and significance are modest.

---

> ### Author Response · Authors · 2020-11-24
> **Response to Reviewer#2**
>
> Thank you for taking your time and effort to take a careful look at our manuscript. We will fix the typos and further polish the paper as you suggest.

---

### Author Response · Authors · 2020-11-24
**General response**

Great thanks for ACs and reviewers taking time and efforts on our work. We will further polish this work by these constructive suggestions.

---

### Decision · Program_Chairs · 2021-01-07
**Final Decision**

**Decision:**

Reject

**Comment:**

This paper proposes a contribution aiming at understanding the cause of errors in few-shot learning. The motivation is interesting but the reviewers pointed out many aspects that require more precisions and polishing in addition to the fact that the upper bound provided it rather loose. The rebuttal provided addresses some concerns, but there are still some remarks that require some clarifications en work.
Hence, I propose rejection.